

# Optimisation of a multi-element airfoil for application to airborne wind energy

Gianluca De Fezza and Sarah Barber

Eastern Switzerland University of Applied Sciences (OST)

Oberseestrasse 10, 8640 Rapperswil, Switzerland

**Correspondence:** Gianluca De Fezza (gldefezza@gmail.com)

**Abstract.** Multi-element airfoils can be used to create high lift, and have previously been investigated for various application such as in commercial airplanes during take-off and landing and in the rear end of Formula One cars. Due to the high lift, they are also expected to have a high potential for application to airborne wind energy (AWE), as confirmed by recent studies. The goal of this work is to investigate and optimise a multi-element airfoil for application to AWE, in order to further the understanding and improve the knowledge base of this high-potential research area. This is done by applying the Computational Fluid Dynamics code OpenFOAM to a multi-element airfoil from the literature (the "baseline"), set up for steady-state 2D simulations with a finite volume mesh generated with snappyHex mesh. Following a grid dependency study and a feasibility study using simulation data from the literature, the angle of attack with the best performance in terms of $E^2 C_L$ ($E$ = glide ratio, $C_L$ = lift coefficient) is identified. The maximum $E^2 C_L$ is found to be approximately seven times larger than that of a typical single-element AWE airfoil, at an angle of attack of 17°. Having found the ideal angle of attack, a geometric optimisation is carried out by altering the relative sizes and angles of the separate airfoil elements, first successively and then using promising combinations. The limits of these changes are set by the structural and manufacturing limitations given by the designers. The results show that $E^2 C_L$ can be increased by up to 46.6% compared to the baseline design. Despite the increased structural and manufacturing challenges, multi-element airfoils are therefore promising for AWE system applications, although further studies on 3D effects and drone-tether interactions, as well as wind tunnel measurements for an improved confidence in the results, are needed.

## 1 Introduction

### 1.1 Introduction to Airborne Wind Energy systems

To address climate change and accelerate the energy transition from fossil fuels, renewable energy generation technologies are required. They have to be reliable, efficient, scalable and provide sufficient low cost energy to satisfy current and future demands. Amongst other novel technologies for producing electricity from renewable sources, a new class of wind energy converters has been conceived under the name of airborne wind energy (AWE) systems. In the late 1970s, Miles L. Loyd had the idea of building a wind generator without a tower, using a flying wing connected to the ground by a tether, much like a





kite (Loyd, 1978). The idea of harnessing wind energy at higher altitudes than conventional wind turbines, up to 600 - 1,000

25 m above the ground, has garnered significant interest in the last ten years. Wind speed generally increases with altitude - at 500-1,000 m above ground, the average wind power density is four times higher than at 50-150 m, where a conventional wind turbine generates typically power. Marvel et al. (2013) estimate that a maximum of 1,800 TW of kinetic power might be produced from winds that blow through the whole atmospheric layer, harvesting wind with both regular turbines and high altitude wind energy converters. This shows that harnessing wind power at elevations beyond the reach of conventional wind turbines

could lead to a breakthrough in wind energy generation.

AWE systems are typically composed of two major components - a ground system and at least one aircraft - that are mechanically linked by tethers. Three different concepts can be distinguished among the various AWE solutions: "Ground-gen systems with fixed ground station", in which mechanical energy is converted into electrical energy on the ground and the ground sta-

35 tion is fixed, "Ground-gen systems with moving ground stations", which include kite-driven vehicles on a track, and "Fly-gen systems", in which the conversion is done directly on the aircraft (Vermillion et al., 2021). Besides the overall concept, the type of flight plays an important role. Different concepts include soft kite designs, rigid wing designs with crosswind motion, auto-gyro concepts and lighter-than-air concepts (Vermillion et al., 2021).

Although a large amount of progress has been made in developing prototypes and demonstrators for these different types of AWE concepts, several challenges still remain. These include challenges related to launch and control strategies (Fagiano and Milanese, 2012), flight dynamics (Cherubini et al., 2015; Vander Lind, 2013; Cherubini, 2012), aerodynamic optimisation (Fagiano and Milanese, 2012), structural optimisation (Lütsch, 2015), tether design (Bosman et al., 2013; Schneiderheinze et al., 2015; Inman and Davis, 2012), and reduction of the flying mass (Argatov et al., 2009; van der Vlugt et al., 2019;

Vander Lind, 2013). This paper focuses on the aerodynamic optimisation.

## 1.2 Multi-element airfoils

Multi-element airfoils can produce significantly higher lift than conventional airfoils due to the high curvature that can be reached compared to single-element airfoils, which are limited in their maximum curvature due to manufacturing constraints (Aiguabella Macau, 2011). As well as this, the flow around multi-element airfoils generally stalls at higher angles of attack

(Ragheb and Selig, 2011). For AWE applications, this could improve manoeuvrability and allow the kite to operate in a wider space, which is a key characteristic of an efficient AWE generator (Fagiano and Milanese, 2012). Moreover, there is a high potential to optimise the performance thanks to the numerous geometrical parameters that can be varied. The number of conceivable configurations for a four-element airfoil can easily be in the billions (Misegades, 1981).

Multi-element airfoils have been studied for various applications extensively. According to the flow characteristics and aerodynamic forces analysis in Vimal Chand et al. (2016), a multi-element airfoil with flaps has higher aerodynamic efficiency than a standard airfoil. Multi-element airfoils are already used in a variety of engineering areas. Mostly they find their use in aircraft





during takeoff and landing stages (Sóbester and Forrester, 2014) or in the automotive industry to increase down-force by the rear wing (Aiguabella Macau, 2011). Several optimisation efforts for the application of multi-element airfoils to the aircraft in-
dustry already started in the 1980s. Along with the increasing demand for faster, more fuel-efficient and more resilient aircraft, the design of multi-element airfoils became more complex. During that time, computer programmes were still in their initial stages and the optimisation process was carried out using an empirical method, e.g. Misegades (1981). A decade later, it was already possible to perform automated optimisation processes. For example, Landman and Britcher (1996) used the computer software LabView for this purpose. The real time first order "method of steepest ascent" algorithm to optimise lift coefficient
($C_L$) versus flap vertical and horizontal position at fixed angle of attack was successfully applied (Fox, 1971). This research also found its application in commercial aviation.

More recent papers on the application of multi-element airfoils to conventional wind turbines were published around the years of 2010, when the worldwide installed wind energy capacities grew rapidly (Dorrell and Lee, 2020). Investigations in-
cluding Ragheb and Selig (2011), Narsipur and Selig (2012) and Ribeiro et al. (2012) showed the potential use and optimisation of multi-element airfoils on conventional wind turbines. This opens up new perspectives in terms of aerodynamic and structural characteristics (Narsipur and Selig, 2012). For instance, thanks to the replacement of blade root airfoils with multi-element airfoils, wind turbine performance in terms of maximum lift-to-drag ratio $C_L/C_{D_{max}}$ was found to be increased by 82%. For these studies. the optimisation criteria was chosen to be the maximum of $C_L/C_{D_{max}}$, where $C_{D_{max}}$ is given by the maximum
value of drag coefficient for all angle of attacks (Ragheb and Selig, 2011).

The high potential of multi-element airfoils to increase the aerodynamic performance of AWE systems has been demonstrated recently in Bauer et al. (2018), which showed that very high $C_L$ of above five can be achieved. As well as this, multi-element airfoils have been used previously by companies such as Makani Power (Vander Lind, 2014). Further research
on the topic would be beneficial for building up a solid knowledge base on the topic in the community and to understand further optimisation possibilities in more detail.

## 1.3   Goal of this work

The goal of this paper is to investigate and optimise multi-element airfoils for application to AWE systems. The work aims to
contribute to the existing knowledge base on the topic, which has been recently initiated. This is done by carrying out 2D steady-state Computational Fluid Dynamics (CFD) simulations of a baseline multi-element airfoil in OpenFOAM for checking the simulation feasibility and then optimising the geometry by varying a range of geometrical parameters. The baseline simulations are shown in Section 2, and the optimisation is discussed in Section 3. The conclusions are drawn in Section 4.



## 2 Baseline Simulations

In this section, the details of the baseline geometry are presented, including the meshing and feasibility studies.

### 2.1 Geometry

For the baseline simulations, an existing multi-element airfoil developed for conventional wind turbine applications called MFFS-018 was used (Ragheb and Selig, 2011). This airfoil was designed by converting the outer geometry of an existing wind turbine airfoil (DU 00-W-401) into several different multi-element airfoils using trial-and-error, and estimating the resulting $C_L$ and $C_D$ using the multi-element airfoil analysis programme MSES (Ragheb and Selig, 2011; Drela, 2007). MSES supports the analysis and design of multi-element airfoils by solving the Euler equations with the finite volume method. This allows to perform computations at transonic Mach numbers, in order to predict transitional separation bubbles, shock waves, trailing edge and shock-induced separation (Drela, 2007). The MFFS-018 airfoil demonstrated the highest $C_L$ and was therefore chosen as the baseline in the present paper. The geometry was obtained for use in this work by manually scanning and digitalising the drawing of the airfoil in Ragheb and Selig (2011) to create a .STEP file. The airfoil is divided into four sub-airfoils (or elements), called Main, Strut, Front Flap and Rear Flap as shown in Fig. 1. The red dashed line shows the overall chord, starting from the leading edge of the Main airfoil and ending at the trailing edge of the Rear Flap. The definition of Angle of Attack (AoA) used in the present work is also marked.

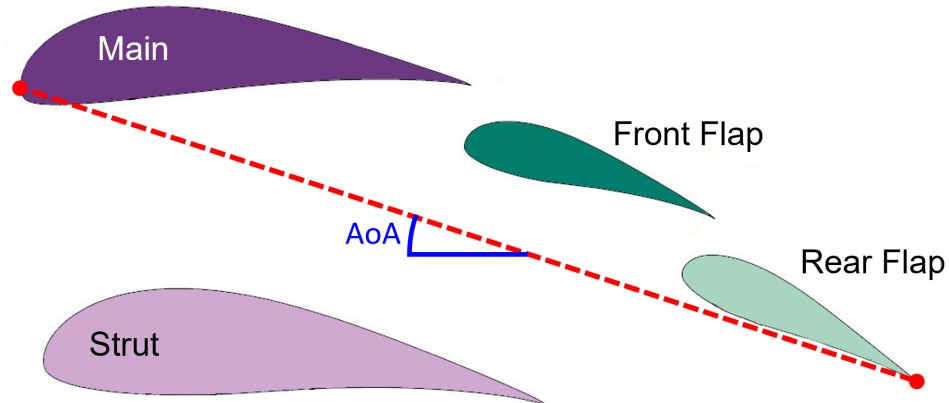

**Figure 1.** MFFS-018 multi-element airfoil baseline geometry (Ragheb and Selig, 2011)

### 2.2 Simulation Set-up

In this work, the CFD code OpenFOAM (Version 6) was applied to the baseline multi-element airfoil, set up for steady-state 2D RANS (Reyonlds Averaged Navier Stockes) simulations. OpenFOAM was chosen due to its ability to capture separated flow with reasonable computational power. It is recognised that 3D effects may have an influence on the results as described in



Bauer et al. (2018), and this is the subject of on-going work.

The mesh, shown in Fig. 2, consists of hexahedrons and was created by the snappyHexMesh utility. By iteratively refining a starting mesh and morphing the resulting split-hex mesh to the surface, the mesh approximates the surface. This allowed to mesh the airfoil in an efficient and automated manner - required later for the optimisation procedure. In the end, each separate airfoil was refined with five mesh layers within the boundary layer, resulting in a domain containing 50,000 cells. This resulted in a y-plus value of 30 with the nutUSpalding wall function (García-Rodríguez and Chacón Velasco, 2020).

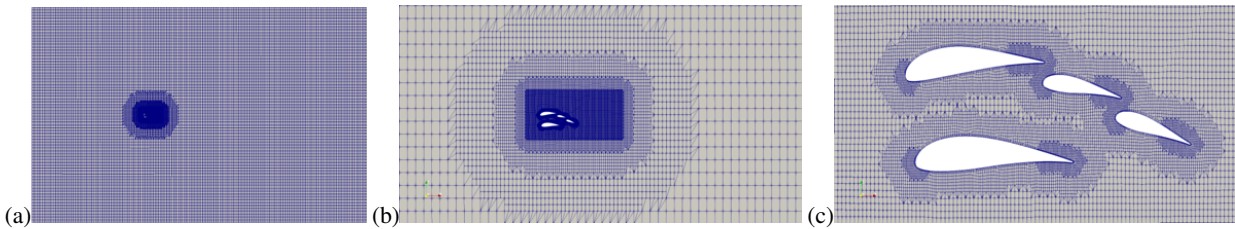

**Figure 2.** (a) CFD domain. (b) Refinement zone around the profile. (c) Refinement layers around the elements.

    The set-up was checked by a grid dependency study, which confirmed that the results of $C_L$ and $C_D$ were not affected by the mesh itself. The results of the mesh dependency study are shown for an AoA of 17° in Fig. 3, where it can be seen that reducing the initial cell size (the size of the cell that was not manipulated by the snappyHexMesh utility) of the domain increased $C_L$ until it became constant below about 20 mm. As well as this, reducing the initial cell size decreased $C_D$ until it became constant

below 20 mm. Studies at other AoAs showed consistent results. Therefore, a initial cell size of 20 mm has been chosen as optimal in this work. The Reynolds' number was chosen to be Re = $10^6$ in order to reflect realistic conditions corresponding to a relative wind speed of 15 m/s and chord length of 1 m and the Spalart-Allmaras turbulence model (Nordanger et al., 2015) was chosen because no significant differences between different turbulence models were found and the one-equation model is comparatively efficient. For this study, it was assumed that the boundary layer was fully turbulent over the entire airfoil due

to the high Reynolds' number applied. Future work could investigate lower-Reynolds' number effects such as boundary layer transition models similar to the work from Folkersma et al. (2019), but this is a secondary priority.

## 2.3   Simulation Feasibility

In order to check the feasibility of the simulations in OpenFOAM, simulations for a range of AoA were compared with MSES simulations from the original study as described in Section 2.1 above (Ragheb and Selig, 2011). A true validation could not be

done because higher-fidelity simulations or wind tunnel measurements are not available to the authors' knowledge. The results are shown in Fig. 4. As a reference length a chord length of 1 m as defined in Fig. 1 was used to calculate $C_L$ and $C_D$. The $C_L$ vs. AoA plot shows that the MSES and the OpenFOAM simulations match fairly well, except around the separation point, as expected. The small offset may be due to different definitions of zero AoA, which was not clear in Ragheb and Selig (2011).





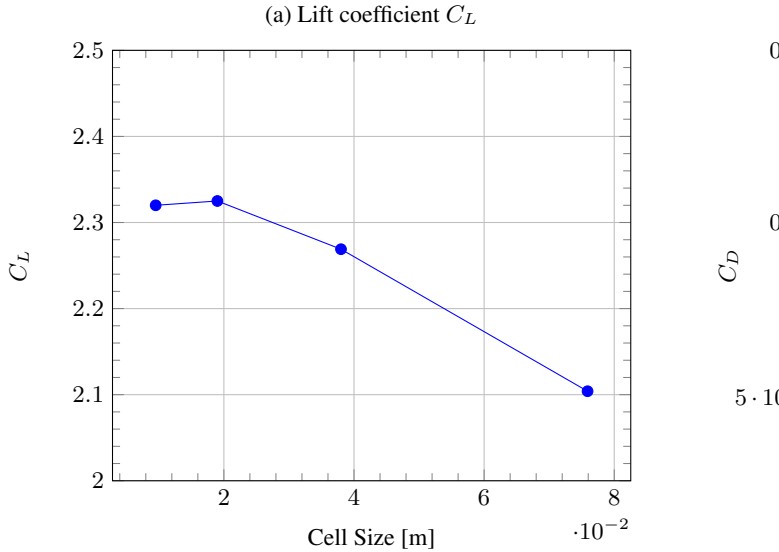

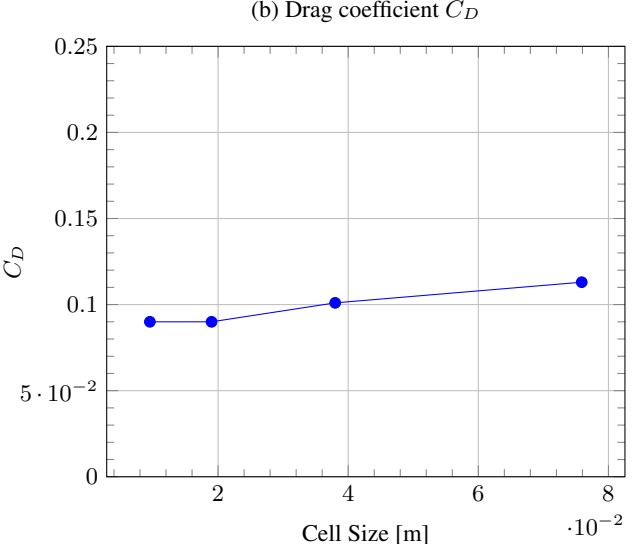

**Figure 3.** Influence of cell sizes on (a) $C_L$ and (b) $C_D$ at AoA = 17°

The $C_D$ vs. AoA plot shows that OpenFOAM over-predicts the drag compared to MSES quite significantly. This is also to be
expected, as CFD takes pressure drag due to flow separation into account (Vinh et al., 1995), whereas MSES does not. Even at
low AoA values, some flow separation can be observed over multiple airfoils (as discussed later in Section 2.4). High-quality
wind tunnel tests are required in order to fully assess the accuracy of both the CFD and the MSES simulations correctly. This
is the topic of on-going work. Thus, with the information available for this study, the set-up was therefore deemed suitable for
further studies, although the absolute values of $C_L$ and $C_D$ should not yet be used directly for AWE designs.

**2.4  Simulation Results**

For this work, instead of $C_L$, the ratio $E^2 C_L$ has been chosen as the optimisation parameter, where $E = C_L/C_D$ = glide ratio
and $C_L$ and $C_D$ are the total lift and drag coefficients of the drone. This variable was chosen since the power of an AWE
system is proportional to $E_{eq}^2 C_L$ (Loyd, 1980), where $C_L$ is the effective system lift coefficient and $E_{eq} = C_L/C_{D_{eq}}$ is the
effective system glide ratio, including the drag of the tether (Bauer et al., 2018). In this work, however, the drag of the tether
has not been yet included, on request of the designers, and $E_{eq} = C_L/C_D$ has been used, where $C_L$ and $C_D$ refer to the total
lift and drag coefficients of the drone, respectively. On-going work involves comparing the results with and without tether drag.

The baseline OpenFOAM results in terms of $E^2 C_L$ are shown in Fig. 5, calculated from the values of $C_L$ and $C_D$ from Fig.
4. It can be seen that the optimum $E^2 C_L$ lies at AoA = 17°.

In order to examine the flow behaviour in more detail, plots of pressure coefficient ($C_p$) versus distance from the Main
leading edge in the main chord direction along the airfoil and streamline visualisations are shown for AoA = 0° in Fig. 6, for



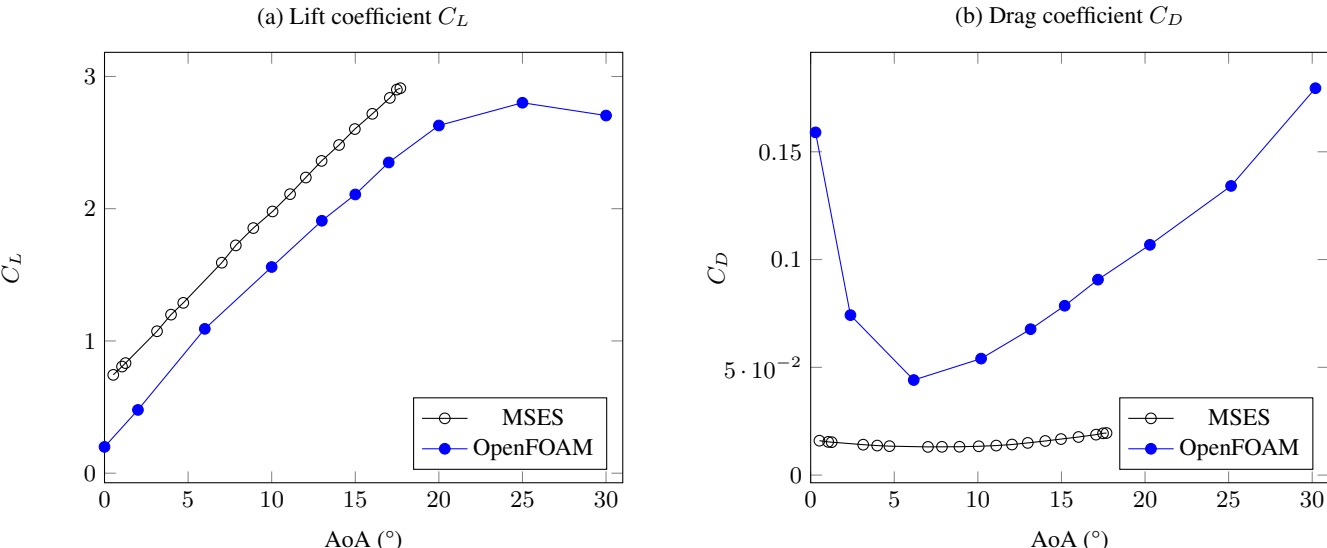

**Figure 4.** Comparison of (a) $C_L$ and (b) $C_D$ with different AoA between MSES and OpenFOAM

AoA = 6° in Fig. 7, for AoA = 10° in Fig. 8, for AoA = 17° in Fig. 9 and for AoA = 25° in Fig. 10. The $C_p$ distribution has been calculated for each airfoil element using the local normal pressure force acting along the surface and the relative chord

length of the individual elements, defined in Fig. 1. The total chord length is 1 m. The x-axis represents the distance from the Main leading edge in the main chord direction along the airfoil.

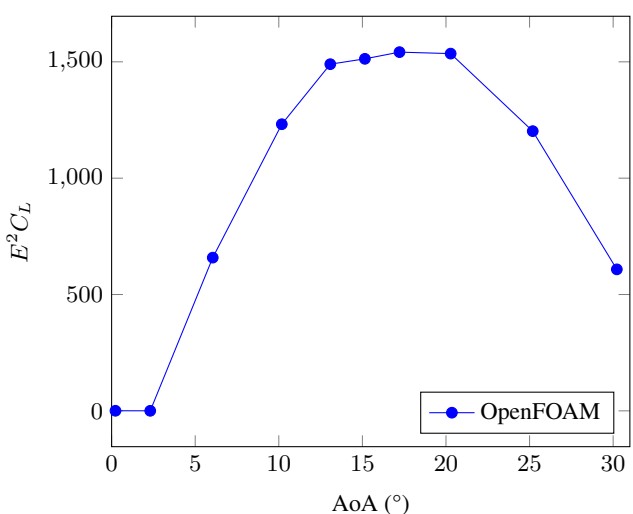

**Figure 5.** Performance of the baseline airfoil in terms of $E^2 C_L$ vs. AoA simulated in OpenFOAM



At AoA = 0°, a substantial pressure-side flow separation can be observed at the Main and Strut elements occurs. The flow
separation is caused by the (local) negative AoA of the Main and Strut elements, although the overall airfoil AoA is zero. Fur-
thermore, a small counter-rotating re-circulation zone at the front of the Front Flap can be seen. The expected counter-rotating
re-circulation zone behind the Strut to match the opposing streamline direction cannot be observed on the streamline plot. Fur-
ther investigations are under-way. However, the $C_p$ distributions of the Front Flap and Rear Flap seem to be mostly unaffected
by flow separation. An increase in AoA to 6° leads so a disappearance of the flow separation, although the $C_p$ distributions for
the Main and Strut elements still indicate small separated regions. It is interesting that such a small change in AoA can lead to
such a significant change in separation behaviour. On-going work is studying this effect in more detail.

The same behaviour can be observed at AoA = 10°. At AoA = 17°, the $C_p$ distribution indicates a lack of flow separation
over all of the airfoil elements. As well as this, they appear to generate the greatest area under the curves, agreeing with the
fact that the $C_L$ is highest at this AoA. Suction side flow separation for the Strut element can be seen at AoA = 25°. This is
visible in the streamline visualisation as well as in the $C_p$ distribution plot. This leads to a decrease in $C_p$.

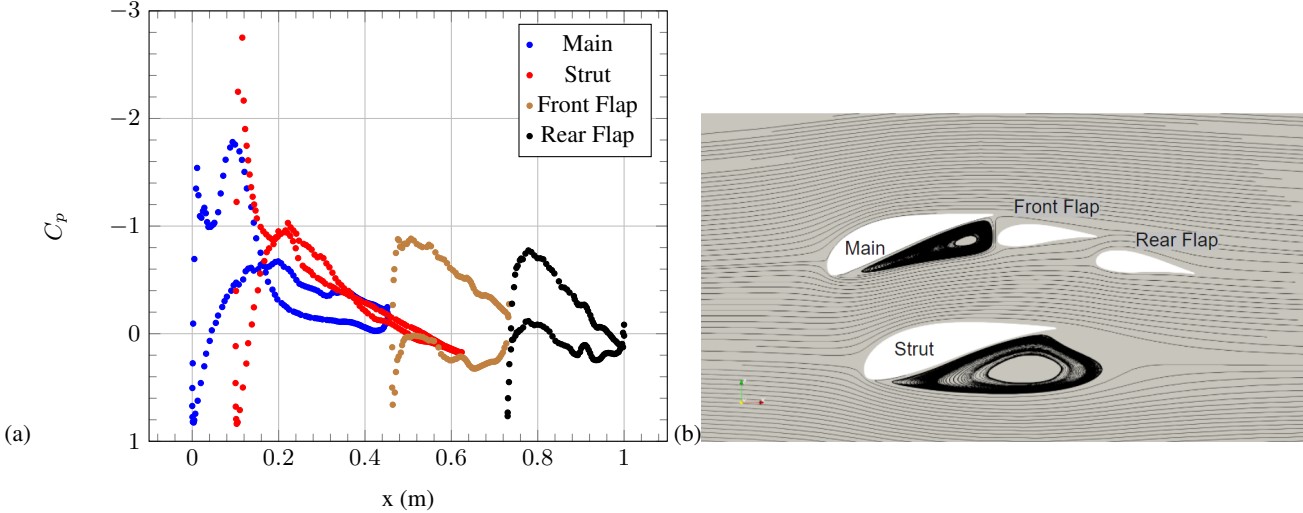

**Figure 6.** (a) Pressure coefficient of the individual element airfoils and (b) streamlines at AoA = 0° (OpenFoam simulations)

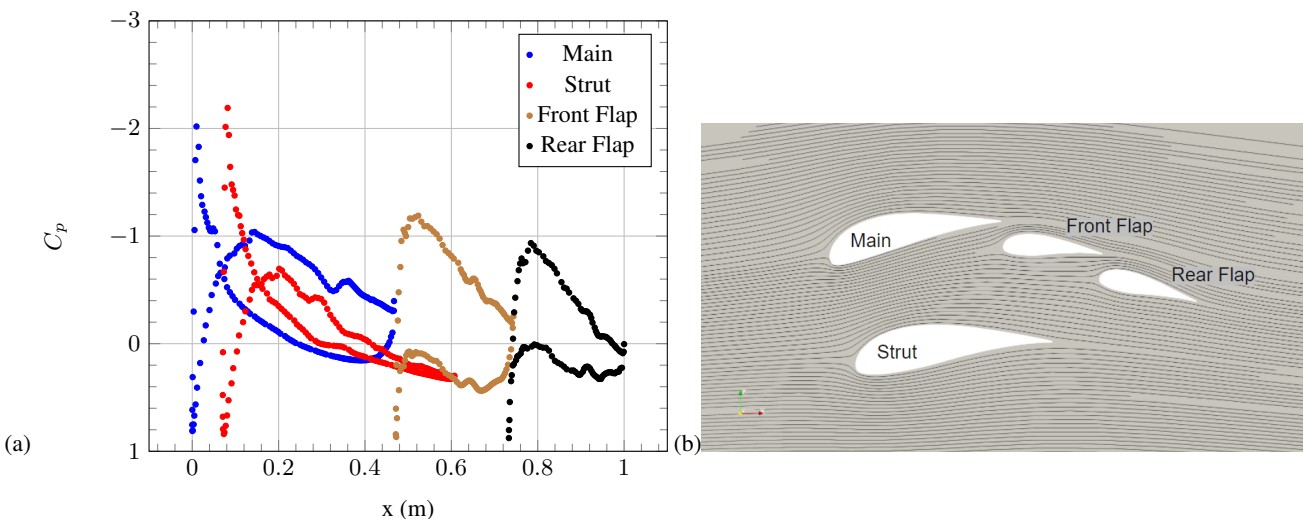

**Figure 7.** (a) Pressure coefficient of the individual element airfoils and (b) streamlines at AoA = 6° (OpenFoam simulations)

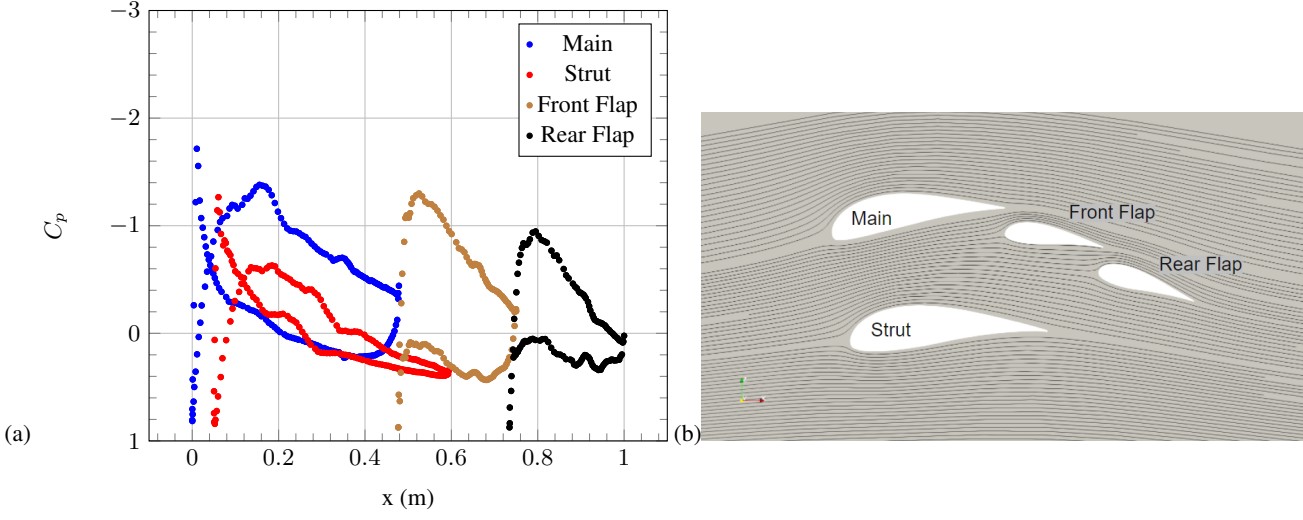

**Figure 8.** (a) Pressure coefficient of the individual element airfoils and (b) streamlines at AoA = 10° (OpenFoam simulations)



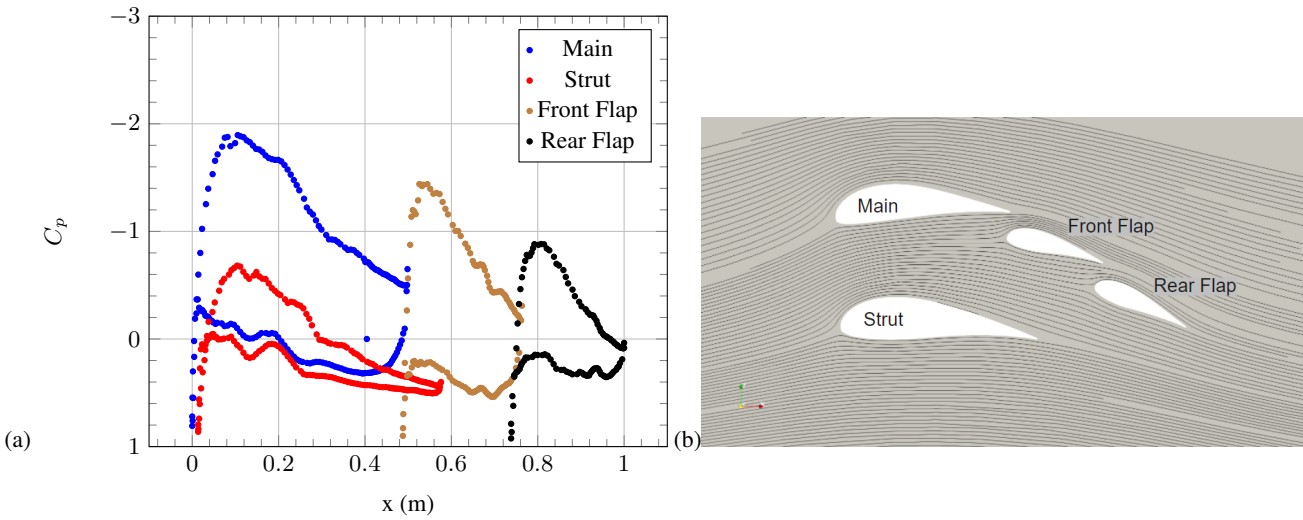

**Figure 9.** (a) Pressure coefficient of the individual element airfoils and (b) streamlines at AoA = 17° (OpenFoam simulations)





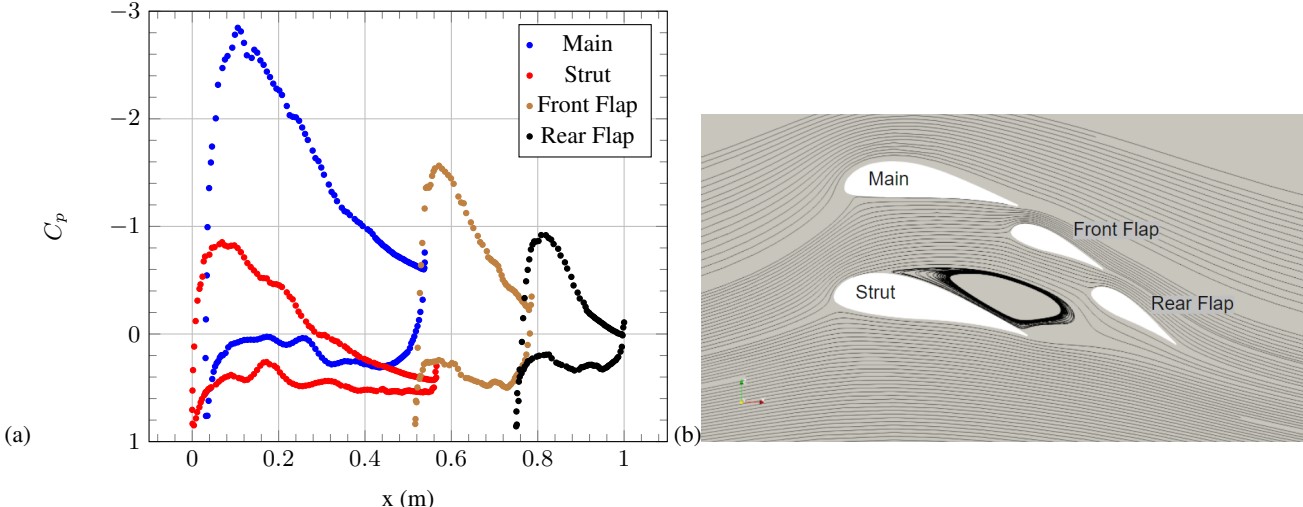

**Figure 10.** (a) Pressure coefficient of the individual element airfoils and (b) streamlines at AoA = 25° (OpenFoam simulations)

## 3 Airfoil Optimisation

In this section the geometric optimisation is carried out. For that, several parameters of the multi-element airfoil were changed and the impact on $E^2 C_L$ quantified. The limits of these changes were set by the structural and manufacturing limits given by the designers.

### 3.1 Optimisation Strategy

For the airfoil optimisation, the effects of certain geometry changes on the airfoil performance were first investigated for the optimal AoA of 17°. For this purpose, individual parameters were changed, simulated and the result in terms of $C_L$, $C_D$ and $E^2 C_L$ compared with the baseline performance. The AoA was not chosen as an input variable in this study in order to reduce the number of simulations carried out, although it is recognised that this could lead to a local maximum being missed and is

180 the topic of on-going work.

The parameters that were investigated are summarised in Table 1. The "Relative Scale" refers to the relative change in size of an individual airfoil element compared to the original geometry in both x- and y-directions equally, i.e. an increase in chord length increases the thickness equally to maintain the shape of the airfoil. A value of 100% refers to no change compared to the

185 baseline. The total chord length of the multi-element airfoil is maintained at all times, meaning that a relative increase in the size of one airfoil leads to a reduction in the size of the others. The "Relative Angle" refers to the rotation in degrees relative to the baseline angle, with an axis of rotation at the leading edge of the element in question. The "Vertical Distance of Strut" refers to the relative distance between the centrelines of the Strut and the Main airfoils compared to the baseline of 100%.



When these parameters were changed, it was taken care that the "overhang" and "gap" between the Main and the Front Flaps as well as between the two flaps remained the same (see Fig. 11). If an AoA variation were to be included in the future, the changes in these parameters due to changing AoA would have to be taken into account. The reference point was always the leading edge of the elements. The ranges and step sizes of the different configurations simulated in this work are also given in the table. The ranges were defined by the designers of the AWE system, due to their structural and manufacturing constraints. The optimisation matrix represents 576 discrete parametric combinations. In order to simulate all these combinations, it would have been beyond the scope of this work and is part of following studies. For this reason, only the influences of the individual changes were assessed.

| Parameter | Range | Step Size |
|---|---|---|
| Scale of Flaps | 60% ... 140% | 20% |
| Scale of Strut | 20% ... 140% | 20% |
| Angle of Flaps | -10° ... +10° | 5° |
| Vertical Distance Strut-Main | 65% ... 275% | 35% |

**Table 1.** Summary of varied parameters; 100% refers to the baseline case

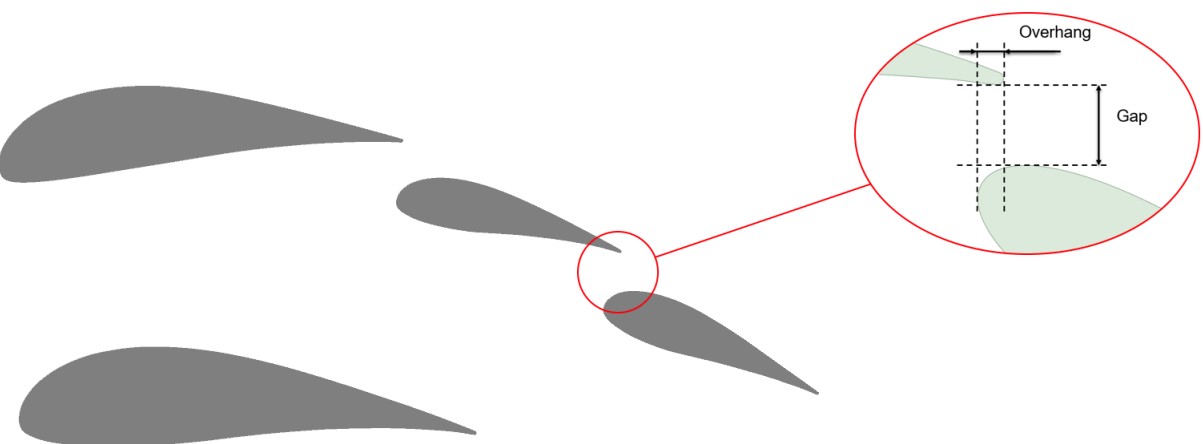

**Figure 11.** Definition of "overhang" and "gap"

For each simulation, the airfoil and mesh were re-created manually in OpenFOAM. Automated optimisation algorithms such as Gradient Descent Method (Ruder, 2016; Zhang et al., 2021) and the Efficient Global Optimization (EGO) algorithm (Jones et al., 1998) were considered, but the manual method was used for this initial study because this approach has the potential to help understand the aerodynamics of multi-element airfoils for this application better. Automated optimisation procedures

 

could be applied in the future, but it is preferable to first have a detailed understanding of the problem. On-going work involves the application and test various optimisation algorithms.

## 3.2 Optimisation Results

Each geometry modification was found to have a different impact on the performance, as discussed in the following sections.

### 3.2.1 Relative Scale of Flaps

Figure 12 shows how the relative scale of the Rear and Front Flaps influence the aerodynamic performance of the airfoil. For each plot, the relative scale of the Front Flap is shown on the x-axis, and the relative scale of the Rear Flap on the y-axis. The colours refer to the absolute values of $C_L$, $C_D$ and $E^2C_L$, respectively. A cubic interpolation between the simulated points 210 spaced every 0.1% is carried out using the Python package "scipy.interpolate". It can be seen that larger relative scales of Front Flap and Rear Flap result in higher $C_L$ and $C_D$ values, leading to a non-linear variation of $E^2C_L$ with a maximum improvement of 7.7% compared to the baseline, at a Front Flap scaling factor of 131% and a Rear Flap scaling factor of 98%. In Fig. 17 at the end of this section a comparison between the baseline geometry (a) and the geometry with the optimal Front Flap scale (b) is shown.

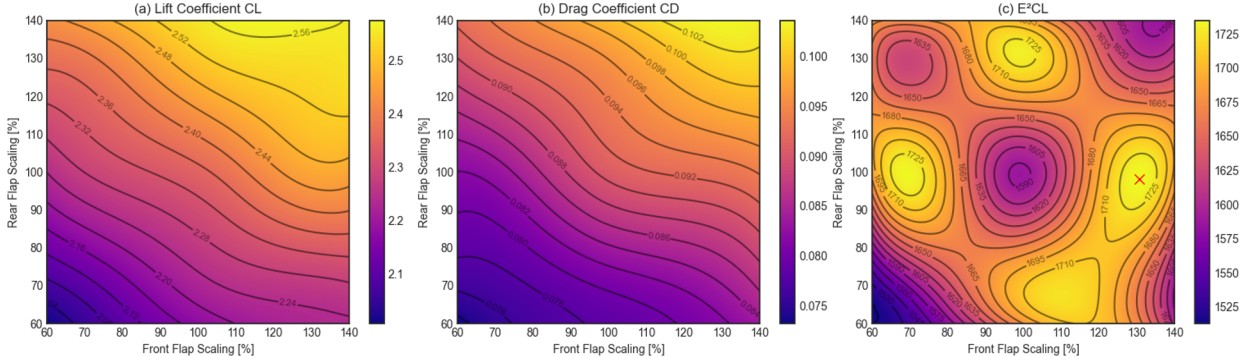

**Figure 12.** Effect of relative scale of Front and Rear Flap on (a) $C_L$, (b) $C_D$ and (c) $E^2C_L$

### 215 3.2.2 Relative Scale of Strut

As shown in Fig. 13, a different effect can be seen when varying the relative scale of the Strut element. Reducing the relative scale of the Strut element leads to higher $C_L$ and only a slightly higher $C_D$. The reason for this is that the flow around the Strut disturbs the flow around the Main profile less if it is relatively smaller. This leads to the fact that $E^2C_L$ can only be improved by decreasing or neglecting the relative scale of the Strut element. Extrapolating this result leads to the conclusion that the Strut 220 element is not beneficial in this aerodynamic system and should be removed unless required for structural reasons. Within the range varied in this study, the smallest relative scale of 20% resulted in an improvement in $E^2C_L$ of 39.9%. In Fig. 17 at the





end of this section a comparison between the baseline geometry (a) and the geometry with the optimal Strut scale (c) can be seen.

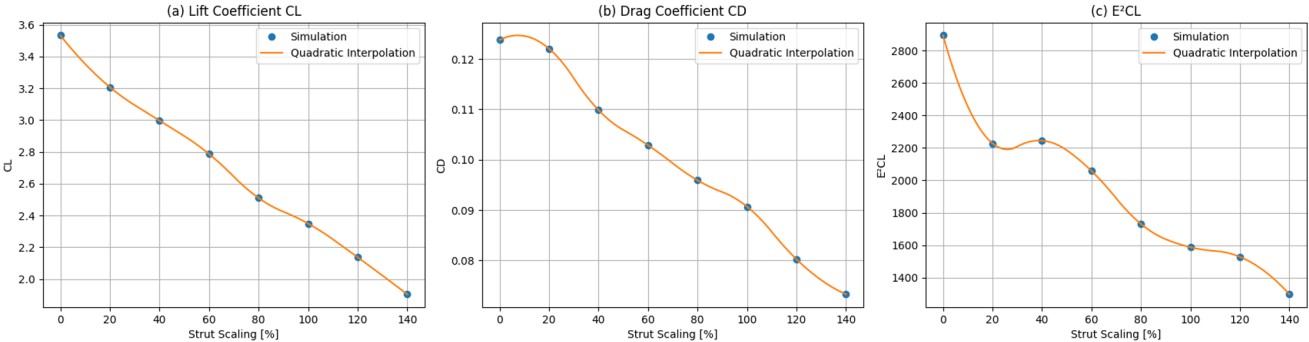

**Figure 13.** Effect of relative Strut scale on (a) $C_L$, (b) $C_D$ and (c) $E^2 C_L$

### 225   3.2.3   Combined Scaling of Flaps and Strut

In order to consider the aerodynamic interactions between the flaps and the Strut element, the Strut, Front Flap and Rear Flap scales were varied together. For that, the Front and Rear Flap were always scaled equally. The results in Fig. 14 show a clear interaction between Flap and Strut scale. In order to achieve the same $C_L$, a smaller Strut was required for small flaps, while a larger Strut was needed for larger flaps. The same applied to $C_D$, but in a different ratio. Therefore, it was not advantageous to make the flaps larger than 120% or the Strut larger than 60%, because above this size the optimal range could no longer by achieved. A maximum improvement in $E^2 C_L$ of 46.6% at a Strut scaling of 43% and a Flaps scaling of 69% could be reached. Due to the mesh inaccuracies, a slight discrepancy occurs between the optimum value of this combination and the sum of the separate optimum values of Strut and flaps. Furthermore, it must be noted that in the combined variant both flaps were simulated with the same scaling. The reason for this is the simplified presentation of the results and the greater impact on the understanding of the geometry changes.

### 3.2.4   Relative Angles of Flaps

In a further step, the relative angle of the Front Flap and Rear Flap was studied. For that, the relative angles of both flaps were varied independently, as shown in Fig. 15. The result shows that Front Flap has to be adjusted less steeply than the Rear Flap in order to improve $E^2 C_L$ by 11.9%. This gives the airfoil a rounder and beneficial aerodynamic shape. In Fig. 17 a comparison between the baseline geometry (a) and the geometry with rotated Rear Flap (d) is shown.





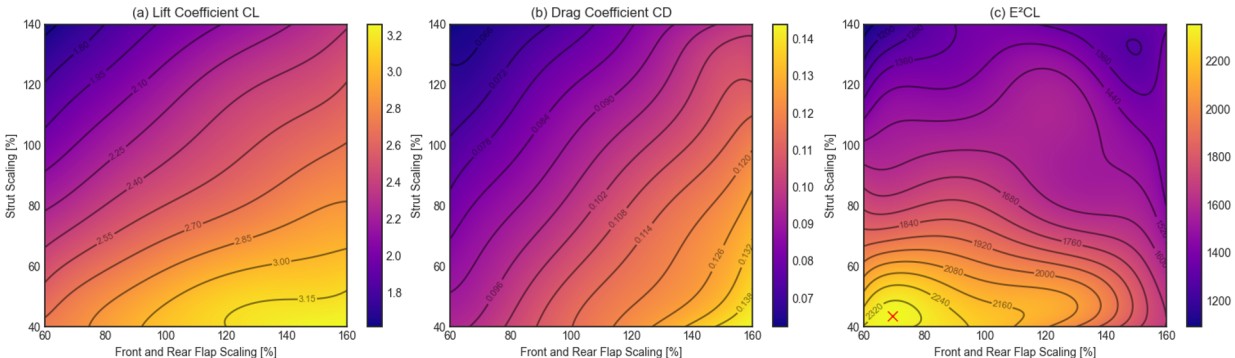

**Figure 14.** Effect of relative Strut and flaps scale altered together on (a) $C_L$, (b) $C_D$ and (c) $E^2 C_L$

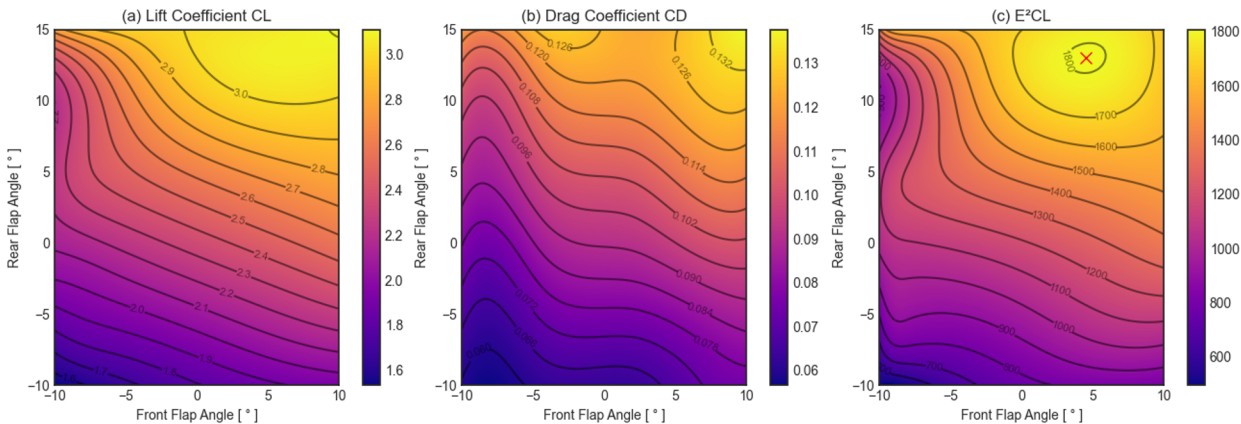

**Figure 15.** Effect of relative Flap angles on (a) $C_L$, (b) $C_D$ and (c) $E^2 C_L$

### 3.2.5 Relative Vertical Distance of Strut

The last optimisation is the modification of the vertical distance of the Strut element relative to the Main element. The results in Fig. 16 show that it is most beneficial to increase the distance between Main and Strut element, meaning to shift the Strut element further down. This result shows once again that the Strut element is aerodynamically not beneficial, since it disturbs the flow around the Main airfoil. It can be seen that the the $E^2 C_L$ value stagnates at higher vertical distances, meaning that no further increase can be achieved by moving it further away. This makes sense, since the Strut element at some point is not part of the aerodynamic system and airfoil anymore and does not disturb the flow. The maximum possible improvement in $E^2 C_L$ is only 7.5%. In Fig. 17 a comparison between the baseline geometry (a) and the geometry with a greater distance between Strut element and Main airfoil (e) is visible.





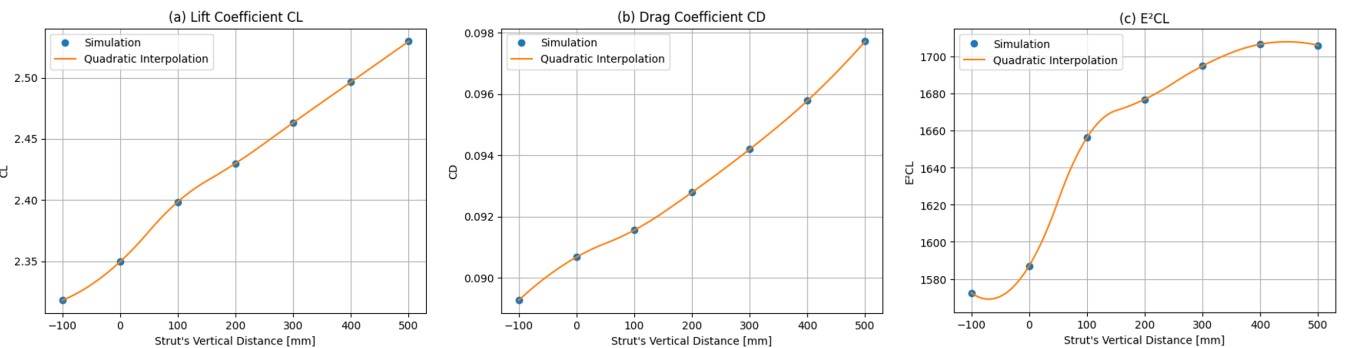

**Figure 16.** Effect of Strut's vertical distance on (a) $C_L$, (b) $C_D$ and (c) $E^2C_L$

## 3.3 Optimal Choice of Geometry

The results above have shown that the optimal geometry for maximising $E^2C_L$ within the given manufacturing and structural constraints has the following properties compared to the baseline:

- Relative Scale of Front Flap = 131%

- Relative Scale of Rear Flap = 98%

- Relative Scale of Strut = 20%

- Relative Angle of Front Flap = 5°

- Relative Angle of Rear Flap = 13°

- Relative Vertical Distance of Strut = 500 mm

The resulting geometry could increase the $E^2C_L$ by 50% in total compared to the baseline geometry. The aerodynamics of the baseline and the optimal geometry can be compared by examining the wake region. For that, four different slices in the wake region were chosen: Position 1 refers to a distance of 5% of chord length ($0.05c$) downstream of the trailing edge of the Rear Flap, Position 2 to $0.1c$, Position 3 to $0.15c$ and Position 4 to $0.2c$. Figure 18 shows the velocity profiles along a vertical line through the wake at these four positions for the baseline (blue) and optimal (orange) geometries. It can be seen that the wake of the baseline geometry has a much lower minimum velocity directly downstream of the trailing edge of the Rear Flap (Position 1). At Positions 2-4, however, the velocity profile of the optimised geometry forms a smooth U-shaped profile more quickly, whereas the baseline geometry still contains the wakes of the separate elements. This indicates that the overall drag is higher. This agrees with previous work - according to Pomeroy (2016), the type of wind speed profile seen here for the optimised geometry indicates better wake properties.

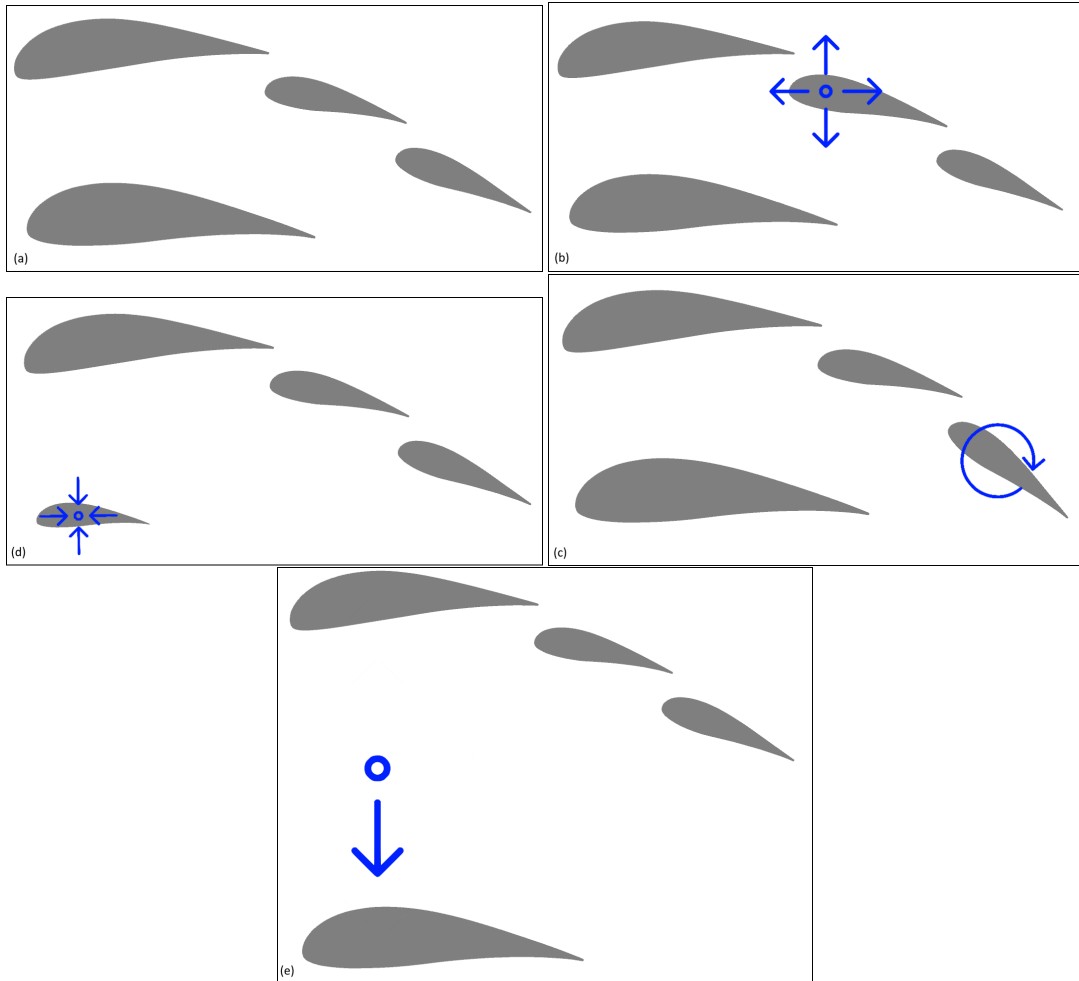

**Figure 17.** (a) Original geometry, (b) Example of geometry with greater Front Flap scale (140%), (c) Example of geometry with steeper Rear Flap angle (-10°), (d) Example of geometry with smaller Strut scale (40%), (e) Example of geometry with greater distance between Main element and Strut (275%).

## 3.4 Suitability of Multi-Element Airfoils for AWE Systems

The results of the optimisation show that multi-element airfoils have a high potential for application to AWE systems. Previous work indicated that the performance (in terms of $E^2C_L$) of AWE systems with multi-element wings can be increased by up to 720% compared to AWE systems with conventional wings (Ragheb and Selig, 2011). The present work used CFD and showed that this geometry could be optimised aerodynamically within the structural and manufacturing constrains by 46.6%, by altering the relative scaling and angles of the individual elements.







**Figure 18.** Velocity profile in wake region at (a) Position 1 (0.05$c$), (b) Position 2 (0.1$c$), (c) Position 3 (0.15$c$) and (d) Position 4 (0.2$c$).

However, the study was limited to 2D CFD and only involved manual optimisation. The consideration of 3D effects will increase the drag and therefore reduce the performance. However, more advanced optimisation methods could help identify some improved optimisation geometries. As well as this, inclusion of the tether drag in the optimisation process is expected to
improve the results.

It should also be noted that despite the improved aerodynamic performance, these types of airfoil could pose some structural and manufacturing difficulties. This study did take into account the limitations of one company, but the details need to be





examined further. For example, the distances between the individual profiles are limited because the space is needed for
production and otherwise there is no room for the tool. Future work could therefore involve taking into account the limitations
connected with the use of manufacturing tools in the optimisation process in the future. Additionally, a coupled aerodynamic
and structural solver would be beneficial, as well as 3D CFD simulations and accompanying wind tunnel tests.

## 4    Conclusions

In this study, the application of multi-element airfoils to AWE systems was investigated. This was done by carrying out 2D
steady-state Computational Fluid Dynamics simulations of a standard multi-element airfoil in OpenFOAM and then optimis-
ing the geometry by varying various geometrical parameters until optimal performance was found. In order to quantify and
optimise the airfoil performance, the term $E^2C_L$ was used, where $E$ = glide ratio and $C_L$ = lift coefficient of the drone.

An existing multi-element airfoil designed for conventional wind turbines was used as the baseline. Following a grid depen-
dency study, baseline simulations were compared to existing simulations using the software MSES, an Euler solver. Although
the lack of wind tunnel data or higher-fidelity simulations did not allow a formal validation to be carried out, this comparison
did confirm the feasibility of the simulations.

For the geometrical optimisation, the optimum angle of attack for the baseline geometry was first identified as 17°. Next,
several geometrical features including the relative scale and angle of the individual airfoil elements were varied separately and
combined in order to identify the most optimal configuration. The constraints were given by manufacturing and structural limits
of the AWE system designer. This brought about significant improvements of 46.6% in $E^2C_L$. Further optimisations would be
possible using automatic optimisation algorithms rather than adjusting the geometry manually. Additionally, an optimisation
strategy that took into account the structural properties and the manufacturing limitations would be beneficial in the future.
Further investigations into 3D effects and tether-drone interactions are on-going.

*Author contributions.* This work was carried out by Gianluca De Fezza as part of his Master's Thesis at the Eastern Switzerland University
of Applied Sciences, and supervised by Sarah Barber, Programme Leader Wind Energy at the Eastern Switzerland University of Applied
Sciences. Gianluca De Fezza carried out all the simulations and analyses, and Sarah Barber managed the project, and contributed significantly
to structuring and writing the paper.

*Competing interests.* The authors declare that they have no conflict of interest.



*Acknowledgements.* This project was part of the Innosuisse project 43730.1 IP-EE "Optimization of a novel Airborne Wind Energy Drone". We would like to thank the Skypull team for their valuable optimisation contstraints and CFD advice (mainly Andrea Pedrioli and Aldo Cattano) as well as our partner research organisation ZHAW (Leonardo Manfriani, Michael Ammann, Marco Caglioti) for carrying out the wind tunnel tests.



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
