# Peer review of "Parameter analysis of a multi-element airfoil for application to airborne wind energy"

_Wind Energy Science, 2021_

## Referee Comment (RC2)

Manuscript ID: WES-2021-155

**Optimisation of a multi-element airfoil for application to airborne wind energy**

Authors: Gianluca De Fezza and Sarah Barber

This work presents an aerodynamic analysis of a multi-element airfoil.

The topic is of definite interest to the scientific community, is well written, and all the points highlighted by the associate editor in his first review have been addressed and implemented by the authors in the current version of the paper.

However in my opinion some changes are needed before the final publication.

**General Comments**

1. The first point I would like to highlight is related to the optimization process. Actually what is presented in Chapter 3 is a (interesting) parametric analysis, where the authors show the effect of different parameters on the aerodynamic coefficient $E^2C_L$. For each of these (or pair of these) parameters, a series of simulations is then made from which the effect on the aerodynamic coefficients is extracted. In these analyses the point of "optimum" is then identified and compared with the baseline (which, I guess is the original industrial configuration). Section 3.3 then combines these individual best conditions to define an optimal geometry on which the wake analysis is then done. This process then does not actually identify the true absolute optimum nor can it be called an optimization process (which instead involves a complete/complex mathematical optimization procedure, in this case constrained, leading to the identification of the optimal parameters and eventually the active constraints). Furthermore, the indicated procedure starts by doing the analyses at an AoA of 17deg, equal to the baseline optimum, which, in general, may not be the final optimum angle of the optimal configuration. However, I think the work is scientifically relevant, so my proposal is to change the title of the paper ("Parameter analysis" instead of "Optimization"). The name optimization in fact induces in the reader an expectation about a mathematical model that is not present in the paper. Moreover, this point and the limitations of this procedure should be emphasized in section 3.3. Line 259/260 are then to be revised since the increase on the aerodynamic coefficient $E^2C_L$ of 50% should be better discussed.

2. Another point is the application of this parametric analysis. The title again refers to AWE system, but, except in section 1 where an extensive analysis of the state of the art of these systems is made, the rest of the paper shows several figures with a 2D model with 4 airfoils. The application to AWE systems should therefore also be strengthened a bit more in the other sections. It is clear that the choice of considering $E^2C_L$ as an aerodynamic parameter (and not E) is already, by itself, something that looks at the world of AWE systems, but probably a figure showing the real system, even a rendering, would help the reader to get immersed in this framework. From this point of view, the often-mentioned construction constraints perhaps should be better defined. In summary, it would be helpful to present in the paper a picture of the baseline system and how this multi-profile is actually realized.

**Comments**

- Lines 74-75. For a wind turbine the aerodynamic parameters to be analyzed is the lift-to-drag ratio $C_L/C_D$. So $C_L/C_{Dmax}$ refers to the best ratio (i.e. best aerodynamic efficiency). So please check the sentence: "[…] where $C_{Dmax}$ is given by the maximum value of drag coefficient[…]".

- Lines 110-120. The final mesh should be better described: 1) the chord length of 1m should be written before (not at line 122), 2) the size of the total domain (figure 2a) should be defined, 3) the cell size of 20mm is the size in which refinement zone? 4) is there some quantitative analysis on the quality of the final mesh?

- Figure 4. The comparison with the literature does not allow a realistic validation of the tool used: the difference in the lift coefficients could be explained by an offset in the AoA, as written, but that of $C_D$, if due to pressure drag, clearly has little meaning. Is there no possibility of adding a further comparison with other data?

- Figs. 6-10. The figures with the pressure coefficients could be improved in their readability if different symbols are used for suction side and pressure side. In so doing, the discussion in the text when referring to the separation in the pressure-side (i.e. line 158) or suction-side (line 169) is easier to be seen in the pictures.

- Fig 6. I'm not sure about the representation of the circular path behind the main/Strut airfoils: I, more realistically, expect a separation zone. I think that here the authors should try to discuss a little bit more this also by commenting on the limits of the 2D steady-state model and/or the effect of the Reynolds.

- Section 3. Already commented in the "general comments"

- Fig 12c. The effect of the Front Flap Scaling seems symmetrical: the optimum value at 130% is very similar to the one at 70%. Why then the choice of this point? What would it change in terms of manufacturing constraints to move in the other direction?

- Figure 18. These are a very interesting analyses. But needs more info. 1) the "y-position on line" (i.e. y-axis) is measured from where? Is the y-origin at the same height of the Strut TE? Maybe a picture with the airfoils may help. 2) Ux in the x-axies has no unit. 3) it's not clear the far field flow velocity value U_inf. Maybe you can plot directly the non-dimensional value Ux/U_inf

**Minor Comments**

- Line 74. "[…]for this studies. the optimization[…]". Check the sentence.

- Line 114. "in a y-plus value of 30 with […]". This is not very clear.

- Lines 121-124. This sentence is too long. Please think of revising it by inserting a full stop for example after "1m".

- Fig 13 and 16. Quadratic interpolation in these figures in some areas has strange trends. Maybe using a "shape-preserving interpolation" method is better [see, for instance fig 16c between -100 and zero or between 100 and 200mm]

---

## Author Comment (AC1)

**General comments**

- Interesting work on using multi-element airfoils for Airborne Wind Energy. While I believe multi-element airfoils can be beneficial for AWE purposes, not much research has been conducted so far in this area. I'm also interested in future work considering optimisation algorithms, including tether drag and 3D effects.
- The figures presented in the paper are clear and add to the understanding of the paper.
- Be a bit more critical of your results and the use of multi-element airfoils in general. Also, discuss possible disadvantages of using multi-element airfoils as well as possible limitations of the models you used. (See specific comments.)

Thank you for taking the time to review this paper in detail. We believe that your specific comments have improved the quality of the paper, and hope that you agree! The changes relevant to your comments are marked in blue in the marked-up manuscript.

**Specific comments**

- Introduction
    - 49: Explain why multi-element airfoils stall at higher angles of attack. How is the flow affected using multi-element airfoils?
    We have explained this better on lines 50-51.

    - 56: It is stated that a multi-element airfoil has a higher aerodynamic efficiency. This cannot be stated in general as this is highly dependent on the metric you are using to assess the efficiency and the specific conditions. Nuance under which conditions and which metric multi-element airfoil can be advantageous.
    Changed, see line 59.

    - 76: Stress out more why exactly multi-element airfoils are beneficial and how they are beneficial for AWE. Multi-element airfoils can reach much higher $C_L$ than single element airfoils, but this comes at a cost of higher $C_D$. To my knowledge, multi-element airfoils do reach higher $C_L/C_D$ than single element airfoils. However, for AWE, power scales with $E^2 C_L$ (Loyd formula), using this metric, $C_L$ is more important, and multi-element airfoils can be beneficial. Moreover, the drag of the tether also adds to the total drag, therefore the drag of the aircraft becomes relatively less important, another argument in favor of multi-element airfoil. I think your research should contribute to validating the above statements.
    Adjusted as suggested (lines 82-84).

- Baseline Simulations
    - 94: At first sight, the presented geometry doesn't look like the DU 00-W-401 airfoil. I had to look up the paper of Ragheb and Selig to understand where the MFFS-018 multi-element airfoil comes from. This is not a very typical multi-element airfoil configuration because of the extra Strut, so I would explain a bit more in detail the choice for this geometry.
    This is already explained in the first paragraph in Section 2.1. We re-read the explanation and aren't really clear how it could be changed so that it is more understandable for you (it makes sense for us!). Could you please let us know if there is anything particular we should change at the start of Section 2.1?

- o 97: transonic Mach numbers are not relevant to your problem.
  No, but this is how the MSES works. We have adjusted the explanation for clarity (line 104).

- o 107: Explain how 3D effects could influence your results.
  We have added an explanation to line 116 (referring to a new figure on request of Reviewer 2).

- o 134: I would say that MSES under-predicts the drag and that OpenFOAM is closer to reality as it is a higher fidelity model.
  Yes, we agree and have changed this sentence (line 153).

- o 145: Comment on how it would influence your results if the drag of the tether is taken into account. Personally, I think it would be very interesting to include this term as it is very relevant for the AWE community to understand how the tether influences the design of an AWE aircraft. Furthermore, it could be relatively easily implemented by assuming a certain drag coefficient, length and diameter of the tether.
  We agree that this would be interesting, but unfortunately the project has finished and it is therefore not within the scope of this work to do this. We are doing it in the next step. We mentioned this already so just made a small change (line 166).

- o 165: I would not consider a change of 6 degrees in AoA small …
  Maybe not. We have reworded this accordingly (line 186).

Airfoil optimisation

- o 271: "multi-element airfoils have a high potential for application to AWE systems." Be specific. On what result is this conclusion based? You did not compare with single element airfoils in your study.

  You are right – the study showed that there is a high potential to significantly improve the performance of existing multi-element airfoils for application to AWE. However, this does imply that they have a high potential for application to AWE, due to the fact that other, non-optimised multi-element airfoils have been compared to single airfoils and found to be suitable. We have reworded lines 312-314 accordingly.

- • Conclusion
  - o I would not call this a conventional wind turbine airfoil, the airfoil is far from conventional.
    On line 329, we have changed "standard" to "existing….from the literature". However, the word "conventional" on line 294 is not referring to the airfoil, but to conventional wind turbines, so we have left it.

**Technical corrections**

- 74: For these studies. the optimization criteria …  remove ". ": changed
- 128: I find "feasibility" a confusing term here, consider using "verification": good point, we have changed this.
- 145: Consider rephrasing "on request of the designer", and explain in more scientific terms why the drag of the tether is not included. We have changed this.
- 156: Main leading edge: not sure what to change here. "Main" is written with a capital "M" because we are referring to the element called "Main". Maybe it's confusing to have the word "main" appearing in small so soon afterwards? ("main chord direction") Anyway we have changed this to "overall" to avoid confusion.
- 239: "rounder and beneficial aerodynamic shape" Vague description, be more precise in explaining results: this is true, we have reworded this.
- 302: Consider: This study has shown that significant improvements up to 46.6% in E2CL are possible. Thank you, we have changed this.

---

## Author Comment (AC2)

Comment on the discrepancies between MSES and CFD results in Figure 4.

When MSES results from *Ragheb and Selig, 2011* are compared with the CFD results, there is an offset in CL and a considerable difference in CD. It is stated that the offset in CL might be because of "different definitions of zero AoA, which was not clear in *Ragheb and Selig (2011)*" (page 5).  And the difference in CD is justified by stating: "as CFD takes pressure drag due to flow separation into account (Vinh et al., 1995), whereas MSES does not." (page 6). Both statements might well be correct although I believe the main reason for discrepancies is not mentioned, which is the fact that the obtained CFD results are with a fully turbulent boundary layer while the MSES results from *Ragheb and Selig, 2011* are obtained with natural transition. I came to this conclusion since I have seen similar results when comparing MSES with CFD-OpenFOAM results for my current work, which also consists on designing and optimizing a multi element airfoil for airborne wind energy applications. According to my work, MSES and CFD will give much closer results when employing the same transition type (forced or natural).

Thank you for this comment. We have adjusted the analysis accordingly in red (see lines 150-151 and 154-156).

---

## Author Comment (AC4)

Thank you for taking the time to thoroughly read and review our paper! We believe that your comments have helped us improve the quality – and hope that you agree! Any changes in the mark-up file are made in green (the other colours are for the other reviewers)

**General Comments**

1. The first point I would like to highlight is related to the optimization process. Actually what is presented in Chapter 3 is a (interesting) parametric analysis, where the authors show the effect of different parameters on the aerodynamic coefficient E2CL. For each of these (or pair of these) parameters, a series of simulations is then made from which the effect on the aerodynamic coefficients is extracted. In these analyses the point of "optimum" is then identified and compared with the baseline (which, I guess is the original industrial configuration). Section 3.3 then combines these individual best conditions to define an optimal geometry on which the wake analysis is then done. This process then does not actually identify the true absolute optimum nor can it be called an optimization process (which instead involves a complete/complex mathematical optimization procedure, in this case constrained, leading to the identification of the optimal parameters and eventually the active constraints). Furthermore, the indicated procedure starts by doing the analyses at an AoA of 17deg, equal to the baseline optimum, which, in general, may not be the final optimum angle of the optimal configuration. However, I think the work is scientifically relevant, so my proposal is to change the title of the paper ("Parameter analysis" instead of "Optimization"). The name optimization in fact induces in the reader an expectation about a mathematical model that is not present in the paper. Moreover, this point and the limitations of this procedure should be emphasized in section 3.3. Line 259/260 are then to be revised since the increase on the aerodynamic coefficient E2 CL of 50% should be better discussed.

We completely agree with this, and actually forgot to change the title after removing the optimisation part of the paper (coming in the future). We have therefore changed the title, as well as changed several other comments, including the abstract, conclusion and section 3.3 (marked in green).

2. Another point is the application of this parametric analysis. The title again refers to AWE system, but, except in section 1 where an extensive analysis of the state of the art of these systems is made, the rest of the paper shows several figures with a 2D model with 4 airfoils. The application to AWE systems should therefore also be strengthened a bit more in the other sections. It is clear that the choice of considering E2CL as an aerodynamic parameter (and not E) is already, by itself, something that looks at the world of AWE systems, but probably a figure showing the real system, even a rendering, would help the reader to get immersed in this framework. From this point of view, the often-mentioned construction constraints perhaps should be better defined. In summary, it would be helpful to present in the paper a picture of the baseline system and how this multi-profile is actually realized.

This is another very good point – thank you! We have included a figure and brief description of the AWE drone to which these results are being applied (section 1.3 and section 3.1).

**Comments**

• Lines 74-75. For a wind turbine the aerodynamic parameters to be analyzed is the lift-to-drag ratio CL/CD. So CL/CDmax refers to the best ratio (i.e. best aerodynamic efficiency). So please

check the sentence: "[…] where CDmax is given by the maximum value of drag coefficient[…]".
Yes, this should be (CL/CD)max. We have adjusted this on line 76.

• Lines 110-120. The final mesh should be better described: 1) the chord length of 1m should be written before (not at line 122), 2) the size of the total domain (figure 2a) should be defined, 3) the cell size of 20mm is the size in which refinement zone? 4) is there some quantitative analysis on the quality of the final mesh?

We have changed all these points (section 2.3), except (4) – the only quantitative analyses are the results of the grid dependency study shown in Figure 3 and the "verification" part in section 2.3. There are no wind tunnel tests for a proper validation.

• Figure 4. The comparison with the literature does not allow a realistic validation of the tool used: the difference in the lift coefficients could be explained by an offset in the AoA, as written, but that of CD, if due to pressure drag, clearly has little meaning. Is there no possibility of adding a further comparison with other data?

There is no data available for us to do a proper validation, which is why we have now changed section 2.3 to "verification" rather than "validation" (on request of Reviewer 1 – see blue changes). Wind tunnel tests are underway, however.

• Figs. 6-10. The figures with the pressure coefficients could be improved in their readability if different symbols are used for suction side and pressure side. In so doing, the discussion in the text when referring to the separation in the pressure-side (i.e. line 158) or suction-side (line 169) is easier to be seen in the pictures.

We have changed the symbols on the suction side to make the plots easier to read.

• Fig 6. I'm not sure about the representation of the circular path behind the main/Strut airfoils: I, more realistically, expect a separation zone. I think that here the authors should try to discuss a little bit more this also by commenting on the limits of the 2D steady-state model and/or the effect of the Reynolds.

On line 181, we refer to this zone as flow separation. Is the "circular path" you refer to not just the usual steady-state representation of a separation zone? We have added a discussion about this at the end of Section 2.4.

• Fig 12c. The effect of the Front Flap Scaling seems symmetrical: the optimum value at 130% is very similar to the one at 70%. Why then the choice of this point? What would it change in terms of manufacturing constraints to move in the other direction?

In this case, the manufacturer chose the larger size due to easier manufacturing; however, the other choice should probably be investigated in the future. We have added a comment about this on line 246.

• Figure 18. These are a very interesting analyses. But needs more info. 1) the "y-position on line" (i.e. y-axis) is measured from where? Is the y-origin at the same height of the Strut TE? Maybe a picture with the airfoils may help. 2) Ux in the x-axies has no unit. 3) it's not clear the far field flow velocity value U_inf. Maybe you can plot directly the non-dimensional value Ux/U_inf

We have explained this better on lines 295-297 and added an extra figure (Figure 19).

Minor Comments

• Line 74. "[…]for this studies. the optimization[…]". Check the sentence.

Corrected

• Line 114. "in a y-plus value of 30 with […]". This is not very clear.

We added an explanation

• Lines 121-124. This sentence is too long. Please think of revising it by inserting a full stop for example after "1m".

Done

• Fig 13 and 16. Quadratic interpolation in these figures in some areas has strange trends. Maybe using a "shape-preserving interpolation" method is better [see, for instance fig 16c between - 100 and zero or between 100 and 200mm]

Yes, we see that now – thanks for pointing that out. However, we can't change it now because the project has finished, but we note that point for future interpolations that we do. We added a comment on line 257.